# Asymptotic temperature of a lossy condensate

**Isabelle Bouchoule[*] and Max Schemmer**

Laboratoire Charles Fabry, Institut d'Optique, CNRS, Université Paris-Saclay,
2 Avenue Augustin Fresnel, 91127 Palaiseau Cedex, France

⋆ isabelle.bouchoule@institutoptique.fr

## Abstract

We monitor the time evolution of the temperature of phononic collective modes in a one-dimensional quasicondensate submitted to losses. At long times the ratio between the temperature and the energy scale $mc^2$, where $m$ is the atomic mass and $c$ the sound velocity takes, within a precision of 20%, an asymptotic value. This asymptotic value is observed while $mc^2$ decreases in time by a factor as large as 2.5. Moreover this ratio is shown to be independent on the loss rate and on the strength of interactions. These results confirm theoretical predictions and the measured stationary ratio is in quantitative agreement with the theoretical calculations.

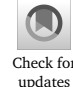

## 1 Introduction

There has been many effort and progress in the last decades for the realization and investigation of isolated many-body quantum systems. The effect of coupling to an environment has however regained interest in the last years. While such a coupling was manly considered as detrimental for the study of many-body quantum physics, it has been shown that

proper engineering of the coupling to an environment could enable the realization of interesting many-body quantum states such as entangled states or highly correlated states [1, 2]. The effect of coupling to an environment is still a widely open question. The simplest kind of coupling, which is ubiquitous in experiments, is a loss process where the particles leave the system. Losses are particularly relevant in exciton-polariton condensates but they are also present, or can be engineered, in ultra-cold atomic degenerate Bose gases. If one considers a Bose-Einstein condensate (BEC) wavefunction, losses are treated as a dissipative term added to the Gross-Pitaevskii equation, equation which describes the evolution of the BEC at the mean-field level. This approach was successful in describing the effect of local losses in an atomic BEC [3] and many aspects of exciton-polariton condensates [4–7]. In the last case, a pumping process ensures the presence of a steady state. Beyond this mean-field approach, the loss process introduces fluctuations, which are due to the shot noise associated to the quantization of the particles. Both the dissipation and the fluctuations produced by losses was taken into account in stochastic theoretical descriptions [8–11][1].

While for exciton-polariton condensates a pumping process is present, in atomic Bose gases the sole effect of losses can be investigated. In [10, 11] the time evolution of a Bose-Einstein condensate, or a quasicondensate in reduced dimension, submitted to homogeneous losses has been theoretically investigated. The dissipative term is responsible for cooling: although the loss process is homogeneous, losses per unit length occur at a higher rate in regions of higher densities – just because there are more atoms – which leads to a decrease of density fluctuations and thus of their associated interaction energy. On the other hand, the stochastic nature of losses tends to increase density fluctuations and thus the interaction energy; this corresponds to a heating term. As a result of the competition between both effects, it has been predicted that phononic collective modes acquire, at large times, a temperature $k_B T$ that decreases proportionally to the energy scale $mc^2$ where $m$ is the atomic mass and $c$ the speed of sound.

The precise value of the asymptotic ratio $k_B T/(mc^2)$ depends on the loss process and the geometry [11]. An intrinsic homogeneous loss process present in cold atoms setup is a three-body loss process where a loss event corresponds to an inelastic collision involving three atoms and amounts to the loss of the three atoms. In [13], the asymptotic ratio $k_B T/(mc^2)$ associated to this three-body losses has been experimentally observed, and its value is in agreement with theoretical predictions. On the other hand, there was up to now no experimental evidence of an asymptotic ratio $k_B T/(mc^2)$ in the case of a one-body loss process [14]. A one-body loss process corresponds to a uniform loss rate: each atom has the probability $\Gamma dt$ to be lost during a time-interval $dt$, regardless both of its position and its energy. In this paper, we demonstrate the presence of an asymptotic value of the ratio $k_B T/(mc^2)$ for one-dimensional harmonically confined quasicondensates submitted to one-body losses, and our results are in agreement with theoretical predictions.

## 2 Description of the experiment and data analysis

We use an atom-chip set-up, described in detail in [15], to produce ultracold gases of $^{87}$Rb atoms, polarized in the stretch state $|F = 2, m_F = 2\rangle$ and confined in a very elongated magnetic trap. The transverse confinement is realized by three parallel wires aligned along $z$, running an AC current modulated at 400 kHz, together with a homogeneous longitudinal magnetic field $B_0 = 2.4$ G [16]: atoms are confined transversely in the time-averaged potential and the transverse oscillation frequency $\omega_\perp/(2\pi)$, which depends on the data-set, lies in the interval [1.5-4.0] kHz. A longitudinal harmonic confinement of frequency $\omega_z/(2\pi) = 9.5$ Hz is realized

---

[1]Other approaches such as the Keldish formalism have been developped [12].

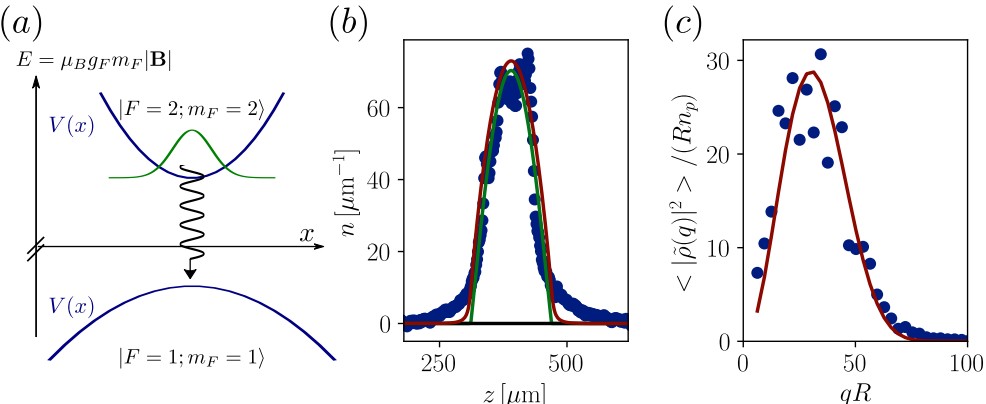

Figure 1: Implementation of one-body losses in our magnetically-confined gas and data analysis. (a) sketch of the MW coupling between the trapped and untrapped state. (b) A typical density profile $n_0(z)$, obtained averaging over about 20 images. The data to which it corresponds (5$^{\text{th}}$ data point of data set 6) is the encircled data point in Fig.(a). The shape expected for a quasicondensate is shown in green, with the peak density $n_p$ as single fitting parameter. (c) Density ripples power spectrum for the same data as (b), together with the theoretical fit yielding the temperature $T$. The red solid line in (b) is the density profile expected for a cloud at a temperature $T$ using the Yang-Yang equation of state and the local density approximation.

by a pair of wires perpendicular to $z$. Using standard radio-frequency (RF) evaporative cooling, we prepare clouds whose temperature $T$ and chemical potential[2] $\mu_p$, depending on the data set, lie in the range $\mu_p/(2\pi\hbar) \in 1.0 - 3.1\,\text{kHz}$ and $T \in 40 - 75\,\text{nK}$. The ratios $k_B T/(\hbar\omega_\perp)$ and $\mu_p/(\hbar\omega_\perp)$ lie in the range $0.3 - 0.7$ and $0.6 - 1.2$ respectively such that the clouds are quasi-one-dimensional. The clouds lie deep in the quasicondensate regime [17], which is characterized by strongly reduced density fluctuations – the two-body zero distance correlation function $g^{(2)}(0)$ being close to one – while longitudinal phase fluctuations are still present. We then increase the frequency of the RF knife by about 25 kHz, so that it no longer induces losses but ensures the removing of residues of three-body recombination events.

In contrast to the three-body process, no intrinsic process leads to a one-body loss term in our experiment and one-body losses have to be engineered. We introduce homogeneous one-body losses by coupling the trapped atoms to the untrapped state $|F = 1, m_F = 1\rangle$, which lies at an energy $\hbar\omega_{HFS} + (3/2)\mu_B B_0$ below the trapped state $|F = 2, m_F = 2\rangle$, where $\omega_{HFS} \simeq 6.8347\,\text{GHz}$ is the hyperfine splitting of the $^{87}$Rb ground state. Coupling is realized by a microwave (MW) field produced by a voltage-controlled oscillator connected to an antenna placed a few centimeter away from the atomic cloud. We use a noise-generator to produce a MW power spectrum which presents a rectangular shape 200 kHz wide. Its central frequency $\omega_0$ may be varied in time. During the preparation phase of our ultra-cold cloud, $\omega_0$ is chosen such that the transition is shifted from resonance by about 5 MHz so that the MW does not induce any noticeable losses. At time $t = 0$, we suddenly shift $\omega_0$ to its resonance value to induce losses. The large width of the MW power spectrum, compared to $\omega_\perp$ and to the chemical potential of the atoms, ensures that the loss rate is homogeneous over the size of the atomic cloud and is not affected by interaction effects. We adapt the loss rate $\Gamma$ adjusting the power of the MW field.

We analyze the atomic cloud using absorption images taken after a time of flight $t_f = 8\,\text{ms}$ following the sudden switch off of the confining potential. We acquire an ensemble of about

---

[2]The energy offset used for chemical potential is the energy of the transverse ground state, *i.e.* $\hbar\omega_\perp$.

20 images taken in the same experimental conditions. The fast transverse expansion of the cloud provides an effective instantaneous switch off of the interactions with respect to the longitudinal motion and the gas evolves as a non-interacting gas for $t_f$. Averaging over the data set, we extract the longitudinal density profile $n_0(z)$. The longitudinal velocity width, of the order of $k_B T/(\hbar n_p)$, where $n_p$ is the peak linear density [18], is small enough so that the longitudinal density profile is not affected by the time-of-flight and $n_0(z)$ is equal to the density profile of the cloud prior to the trap release. From $n_0(z)$, we extract the total atom number and the peak density $n_p$. The latter is obtained by fitting the central part of the measured density profile with the profile expected for a gas lying in the quasicondensate regime. To compute the quasicondensate density profile we rely on the local density approximation (LDA): the gas at position $z$ is described by a homogeneous gas at chemical potential $\mu(z) = \mu_p - m\omega_z^2 z^2/2$, and the linear density is derived from $\mu(z)$ using the equation of state of a homogeneous quasicondensate. The latter, which relies the chemical potential $\mu$ to the linear density $n$, is $\mu = \hbar\omega_\perp(\sqrt{1 + 4na_{3D}} - 1)$, where $a_{3D}$ is the 3D scattering length [19, 20]. For $na_{3D} \ll 1$ it reduces to the pure 1D expression $\mu = g_{1D}n$ , where the 1D coupling constant[3] is $g_{1D} = 2\hbar\omega_\perp a_{3D}$ [21]. At larger $na_{3D}$ it includes the effect due to the transverse swelling of the wavefunction. The longitudinal quasicondensate profile extends over $2R$, where $R = \sqrt{2\mu_p/(m\omega_z^2)}$. Fig. (1)(b) shows a typical experimental density profile $n_0$, together with the theoretical quasicondensate profile. The good agreement between most of the cloud's shape and the calculated profile confirms that the cloud lies deep into the quasicondensate regime. It also confirms that the loss rate is small enough so that the cloud shape has time to follow adiabatically the atom number decrease.

Temperature determination is realized by the well-established density-ripple thermometry method [13, 22–25]. This thermometry uses the fact that thermally excited phase fluctuations initially present in the cloud transform into density fluctuations during $t_f$ such that single shot images of the cloud presents large random density ripples. From the set of acquired images, we extract the power spectrum of the density ripples. More precisely, we extract from each image $\rho_q = \int_{-R}^{R} dz \delta n(z)e^{iqz}$ where $\delta n(z) = n(z) - n_0(z)$ and we then compute the density ripple power spectrum $\langle |\rho_q|^2 \rangle$, from which we remove the expected flat-noise contribution of optical shot noise. The power spectrum is then fitted with the expected power spectrum for a quasicondensate of peak density $n_p$ confined in a harmonic longitudinal potential, calculated using the LDA approximation [25], with the temperature as fitting parameter[4]. Fig. 1(c) shows an example of a power spectrum (corresponding to the encircled data point in Fig. (2)(a), together with the theoretical fit yielding the temperature $T$. This thermometry probes fluctuations whose wavelengths are much larger than the healing length $\xi = \hbar/\sqrt{mgn_p}$ such that the temperature corresponds to the temperature of the phononic collective modes.

## 3   Experimental results

We investigate the time evolution of the atomic cloud for 6 different data sets which correspond to different transverse oscillation frequencies – *i.e.* different interaction 1D effective coupling constant –, different initial situations – *i.e.* different atom number and temperature – and different MW power – *i.e.* different 1-body loss rate. They are listed in table (1).

We plot in Fig. (2)(a) the time evolution of the total atom number for the different data

---

[3]In our case, $\omega_\perp \ll \hbar/(ma_{3D}^2)$ such that we are far from confinement-induced resonance.

[4] We take into account the finite imaging resolution by multiplying the theoretical power spectrum with $e^{-k^2\sigma^2}$ where $\sigma$ is the rms width of the imaging point-spread-function. Due to finite depth-of-focus, $\sigma$ depends on the size of the cloud along the imaging axis, which itself depends on $\omega_\perp$. Thus $\sigma$ may depend on the data set but for a given data set we use the same $\sigma$ for all evolution times.

Table 1: Data sets presented in this paper, with the associated symbol used in the figures 2 and 3.

| data-set number | $\omega_\perp/(2\pi)$ (kHz) | $\Gamma$ (s$^{-1}$) | Symbol |
|---|---|---|---|
| 1 | 1.5 | 3.8 | ◀ |
| 2 | 1.5 | 1.6 | ★ |
| 3 | 2.1 | 5.2 | ● |
| 4 | 3.1 | 4.9 | ■ |
| 5 | 3.1 | 2.5 | ▲ |
| 6 | 4.0 | 4.5 | ▶ |

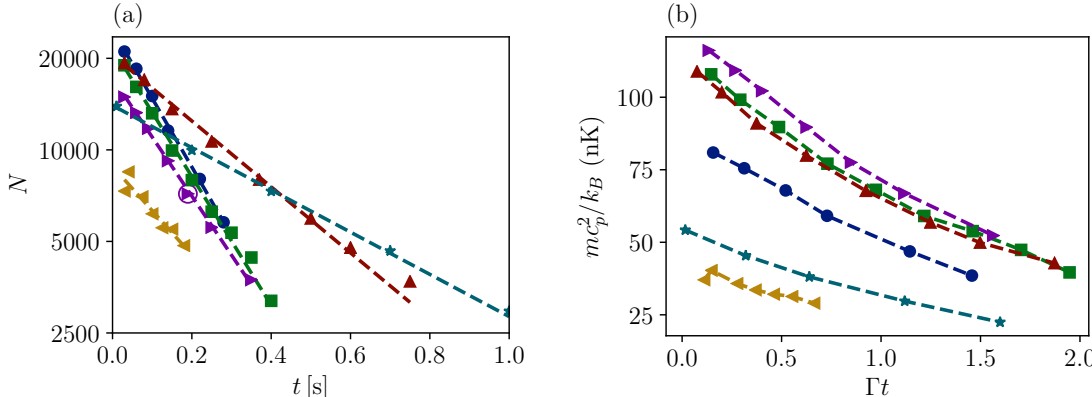

Figure 2: (a) Evolution of the total atom number for the different sets of data shown in semi-log scale. The loss rate $\Gamma$ for each data set is deduced from an expontential fit, shown as dashed line. (b) Evolution of the energy scale $mc_p^2$ for same data sets as a function of $\Gamma t$. Both plots use the color codes of table. 1

sets. The exponential decrease of the atom number, shown by the good agreement with exponential laws represented by straight line in the semi-log plot, confirms that we realized a uniform one-body loss process. The loss rate varies by roughly a factor 2 between different data sets. As pointed out in the introduction, a relevant energy scale is $mc_p^2$ where $c_p$ is the sound velocity computed at the center of the cloud, which fulfills $mc_p^2 = n\partial_n\mu|_{n=n_p}$. For pure 1D quasicondensates, $mc_p^2 = g_{1D}n_p$ where $n_p$ is the peak density. In our data sets the linear densities can reach values which are not small compared to $1/a_{3D}$ and we use the more general expression $mc_p^2 = n_p g_{1D}/\sqrt{1 + 4n_p a_{3D}}$. The evolution of the energy $mc_p^2$ for the data sets is shown in Fig. (2)(b). The variation of $mc_p^2$ during time is as large as a factor 2.5.

The time evolution of the ratio $y = k_B T/(mc_p^2)$ is shown in Fig. (3)(a) for all the data sets. Theory for 1D harmonically confined gases [11] predicts that $y$ converge at long times towards the asymptotic value $y_\infty^{\text{theo}} = 0.75$, shown as solid black line in Fig. (3)(a). The observed behavior is compatible with this prediction: the spread of values of $y$ among different data sets decreases as $\Gamma t$ increases and at long times, all data gather around $y_\infty^{\text{theo}} = 0.75$, regardless of the loss rate $\Gamma$ and of the transverse oscillation frequency. For the data sets 2,3 and 6, $y$ deviates by no more than 20% from $y_\infty^{theo}$ over the whole time evolution while $mc_p^2$ decreases by a factor up to 2.5. For all data sets $y$ is about stationnary for times $t > 0.7/\Gamma$ and we note $y_\infty$ the mean value of $y$ for times $t > 0.7/\Gamma$. Fig. (3)(c) shows $y_\infty$, plotted versus the transverse oscillation frequency. Results are close to $y_\infty^{theo}$, with $|y_\infty - y_\infty^{theo}|/y_\infty^{theo} < 0.2$. The observed discrepancy between $y_\infty$ and $y_\infty^{theo}$ may be due on the one hand to our finite

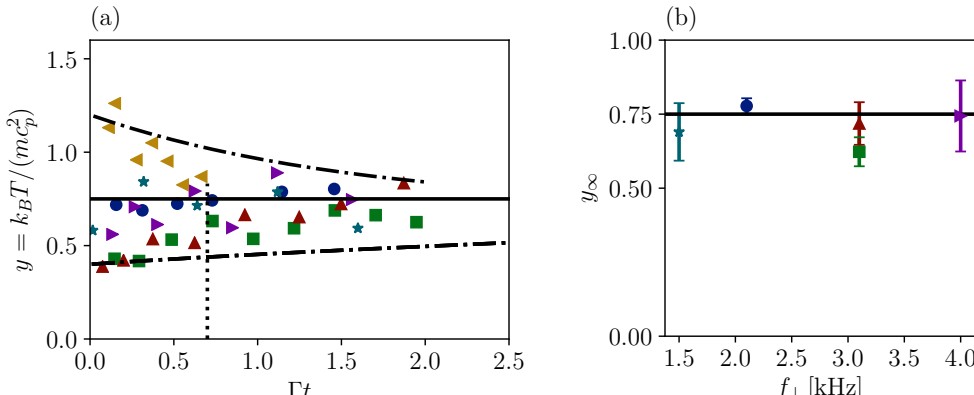

Figure 3: (a) Time evolution of the ratio $y = k_B T/(mc_p^2)$ for the different data sets, shown as a function of $\Gamma t$. The solid horizontal line shows $y_\infty^{theo} = 0.75$. The dashed-dotted lines are the computed expected time evolutions corresponding to initial situations of the data set 1 and 5. (b) For each data set, when available, mean value of $y$ for times larger than $0.7/\Gamma$. Error bars show the standard deviation among the data points that fulfill $\Gamma t > 0.7$.

thermometry precision (the value $y_\infty$ is within one error bar of $y_\infty$ for most data sets) and on the other hand to the fact that the criteria $\Gamma t > 0.7$ might be insufficient to unsure the asymptotic value of $y$ has be attained.

Quantitative experimental investigation of the time-evolution of $y$ under the effect of losses is difficult with our data sets. The reason is that the initial condition we produce are such that the maximal deviation between $y(t = 0)$ and $y_\infty^{theo}$ is comparable to our thermometry resolution. We attribute this to the preparation scheme where, for our experimental procedure, three-body losses during the evaporative cooling probably impose a value of $y$ close to 0.75 [13]. On the theoretical side, for a given initial condition, the expected time evolution of $y$ can be computed using the dynamical equations derived in [11]. For pure 1D harmonically confined cloud the equation reduces to $dy/d(\Gamma t) = y/3 + 1/4$. To take into account the 3D effect due to transverse swelling of the wave-function, we solved numerically the general equations given in [11]. We show in dashed-dotted black lines in Fig. (3)(a) the expected time-evolution for initial conditions corresponding to the 1st and the 5th data sets. The expected convergence of $y$ towards $y_\infty^{theo}$ is found to be very slow for the data set 5. Here transverse swelling effects slow down the dynamics[5]. Experimentally, the convergence appears to be slightly faster. For initial situations corresponding to the data set 1 on the other hand, transverse swelling effects are expected to speed up the dynamics. Data are consistent with this behavior.

## 4 Conclusion

In this paper, we identify for the first time the asymptotic temperature of a 1D quasicondensate submitted to a 1-body loss process: more precisely, we show that the ratio $k_B T/(mc_p^2)$ reaches an asymptotic value, close to the theoretical prediction of 0.75. In a previous work [14] which investigates the evolution of the temperature of a quasicondensate under the effect of losses, 1D quasicondensates were shown to reach lower ratios $k_B T/(mc_p^2)$, in disagreement with the-

---

[5]Because of transverse swelling effect at large density, for some initial parameters, the function $y(t)$ could even be not monotonous.

oretical predictions. The difference between the work [14] and the present work is two-fold. First, in [14] the out-coupling is realised with a monochromatic field, in which case, for chemical potential of the order of the transverse trapping frequency, homogeneity of the loss process is not guaranteed. Such an inhomogeneity makes the loss process sensitive to the energy of the atoms; a phenomena not accounted for by the model. In this paper the use of a wide MW power spectrum ensures the homogeneity of the losses. Second, the data sets in [14] for which ratios $y$ lower than expected are reported, correspond to losses engineered via a radio-frequency field that couples magnetic states within the hyperfine level $F = 2$: in opposition to what happens when using microwave outcoupling to $F = 1$, the transfer of the trapped atoms, which are in the $|F = 2, m_F = 2\rangle$ state, to untrapped states $|F = 2, m'_F \leq 0\rangle$ involves the intermediate state $|F = 2, m_F = 1\rangle$ which is held in the magnetic trap. Since both states contribute to the images and a priori host uncorrelated fluctuations, one expects a decrease of the density ripple power spectrum and thus of the fitted temperature.

This work leads to many open-questions. First, the thermometry we use probes the collective modes which lie in the phononic regime[6], while theoretical predictions [26] indicate that collective modes of higher frequency reach, under the effect of losses, higher temperatures. As already pointed out in [26], for clouds confined in a smoothly varying potential, information on higher frequency collective modes may be retrieved from the wings of the cloud, namely the part of the density profile that extends beyond the size of $R$ of a quasicondensate. Indeed, as losses occur, we observe the growing of the fraction of atoms present in the wings and the density in the wings typically largely exceed that expected for a cloud at thermal equilibrium at a temperature equal to the temperature extracted from the density ripple thermometry, as is shown in Fig. (1). This call for further theoretical and experimental investigations. Second, the theoretical prediction that for phonons the ratio $k_B T/(mc^2)$ reaches an asymptotic value is *a priori* also valid in higher dimensions. It is an open question whether coupling to higher frequency collective modes, an inefficient process in 1D [26], prevents the phonon modes to attain this asymptotic behavior. Finally, it would be interesting to extend the investigation of the effect of losses to regimes different from the (quasi-)condensate regime. In the case of 1D Bose gases with contact interactions, that are described by the Lieb-Liniger model, a description in terms of the evolution of the distribution of rapidities [27] would permit to generalize the studies to all possible states of the gas.

## 5 Acknowledgments

The authors thanks Bernhard Rauer for interesting discussions and Marc Cheneau for his suggestion of using micro-wave field to induce losses. M. S. gratefully acknowledges support by the Studienstiftung des deutschen Volkes. This work was supported by Région Île de France (DIM NanoK, Atocirc project). The authors thank Sophie Bouchoule of C2N (centre nanosciences et nanotechnologies, CNRS / UPSUD, Marcoussis, France) for the development and microfabrication of the atom chip. Alan Durnez and Abdelmounaim Harouri of C2N are acknowledged for their technical support. C2N laboratory is a member of RENATECH, the French national network of large facilities for micronanotechnology.

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
