# Peer review of "Asymptotic temperature of a lossy condensate"

_SciPost Physics, doi:SciPost Phys. 8, 060 (2020)_

## Round 1 · Referee Report · Anonymous (Referee 1) · 2020-2-21

Strengths

1- Very clear introduction 2- Clear description of the data analysis

Weaknesses

1- The major weakness I see is that, since the main quantity addressed, i.e. the ratio $y=k_B T/(mc_p^2)$, does not change at all, it bears no clear evidence of reaching equilibrium

Report

The manuscript reports on the experimental investigation of the equilibrium properties of a one-dimensional quasi Bose-Einstein condensate subject to one-body losses. The one-body losses are obtained by exposing the quasi-BEC to a microwave radiation coupling atoms out of the magnetic trap. Importantly, the microwave spectrum is sufficiently broad to make the losses homogeneous across the sample and independent of the energy of the removed atoms. The temperature of the quasi-BEC is obtained by the analysis of the density fluctuations in TOF images and it is related to the (peak) sound speed $c_p$, as obtained from the peak density. The results show that the sound speed $c_p$ decreases (as expected) over time when the lossed induced by the microwave radiation are present, and that the ratio $k_B T/mc_p^2$ is consistent with the theoretical equilibrium value 0.75. Actually, as duly acknowledged, the ratio is practically constant over, likely because the sample preparation leads to approximately the same value 0.75.

The experiment is clearly described, as well as the data analysis, still I have some remarks that I would like to be clarified. 1) I miss the difference between the densities $n_0(z)$ and $n_{qBEC}(z)$. How is the latter derived from the former? 2) The asymptotic value $y_\infty$ is defined as the mean of values for $\Gamma t>0.7$, why 0.7? 3) I find counter-intuitive that losses, although flat vs energy, lead to a cooling of the gas, it conflicts with the fact that the energy per particle should stay constant. I encourage the Authors to provide some insight on the reason of the observed cooling.

That said, I find the work solid, the results convincing and the limitations clearly described. The manuscript is well written with a clear introduction to the topic. I recommend publication.

Below are listed other minor remarks that the Authors might want to consider: 1-end of page 3: the relation between chemical potential and linear density $\mu = \hbar \omega_\perp \sqrt{\dots}$ should be backed by a Reference; 2- page 5, line 10 from bottom: "Results for are close to \dots", the word "for" should be removed; 3- page 7, line 5: "holded" to be replaced by "held".

Requested changes

1-end of page 3: the relation between chemical potential and linear density $\mu = \hbar \omega_\perp \sqrt{\dots}$ should be backed by a Reference; 2- page 5, line 10 from bottom: "Results for are close to \dots", the word "for" should be removed; 3- page 7, line 5: "holded" to be replaced by "held".

---

## Round 2 · Author Response

We thank a lot the referee for her/his work on our manuscript and for her/his comments and remarks that enable us to improve the paper. We answer below to all comments of the referee.

1) We agree that introducing the notation $n_{qBEC}$ for the calculated theoretical profile of a quasi-BEC was not necessary and even maybe confusing. We no longer introduce this notation.

2) The factor 0.7 we choose to extract $y_{\infty}$ from our data is somehow arbitrary. For times $t>0.7\Gamma$, each data set reached, within our measurement precision, a stationary value. For smaller times, and for some data set (data set 1 and 5), we exclude, with this criteria, data points that might show temporal evolution. We added a sentence in the text: “For all data sets $y$ is about stationary for times $t>0.7/\Gamma$ and we note $y_\infty$ the mean value of $y$ for times $t>0.7/\Gamma$.”

3) Indeed the energy independent loss process is counter-intuitive. The loss rate per atom does not depend on the position of the atom, its potential energy nor the density it sees. However, in presence of repulsive interactions between atoms, this does not imply that the energy per atom – for the remaining atoms-- stays constant during the loss process. First, as the mean density decreases, the interaction energy of the remaining atoms decreases. This however is not sufficient to explain the cooling of the phonon modes. To understand the origin of the cooling, let us consider a density fluctuation $\delta n(z_i)$ at position $z_i$ which is larger than the average density $n_0$: the loss rate per unit length at $z_i$ – equal to $n(z_i) \Gamma$ – is larger than the loss rate per unit length in a zone where the density is $n_0$. In other words, if one considers a lost atom, it has more chance to come from a region of high density than from a region of low density. Thus losses tend to decrease density fluctuations. Since the interaction energy is higher in zone of high densities that in zone of low density, the energy associated to density fluctuations, has decreases: this is at the origin of the cooling of the phononic modes.

In the introduction, we added the following sentences to give a hint of the origin of cooling : “The dissipative term is responsible for a cooling: although the loss process is homogeneous, losses per unit length occur at a higher rate in regions of higher densities-- just because there are more atoms-- which leads to a decrease of density fluctuations and thus of their associated interaction energy. On the other hand, the stochastic nature of losses tends to increase density fluctuations and thus the interaction energy; this corresponds to a heating term. As a result of the competition between both effects, it has been predicted that phononic collective modes at large times acquire a temperature $k_B T$ that decreases in proportionally to the energy scale $mc^2$ where $m$ is the atomic mass and $c$ the speed of sound.”

Requested changes: we made all requested changes.

---

## Round 2 · List of Changes

List of changes :
-Second paragraph of the introduction : We added a sentence to give a hint on the --counter-intuitive-- cooling mechanism
-Page 3, in the description of the analysis of the mean density profile : we remove the notation $n_{qBEC}$
-En of page 3 : we added references for the formula \mu=\hbar\omega_\perp(\sqrt{...)
-Page 5, introduction of $y_{\infty}$ : We added a sentence to justify the introduction of a minimum time $t\Gamma$
-We corrected the typos and orthographic errors pointed out by the referee

---

## Editorial Decision

published